## ⦿ PLOS | ONE

# High prevalence of off-label and unlicensed paediatric prescribing in a hospital in Indonesia during the period Aug.—Oct. 2014

**Brechkerts Lieske Angruni Tukayo**[1,2], **Bruce Sunderland**[1], **Richard Parsons**[3], **Petra Czarniak**[1] *

**1** School of Pharmacy and Biomedical Sciences, Curtin University, Bentley, Western Australia, Australia, **2** Health Polytechnic Jayapura Indonesian Ministry of Health, Jakarta, Indonesia, **3** School of Occupational Therapy, Social Work and Speech Pathology, Curtin University, Bentley, Western Australia, Australia

* P.Czarniak@curtin.edu.au

## Abstract

### Background

Data on off-label and unlicensed prescribing in children in Indonesia is limited. The aims of this study were to determine the prevalence of off-label and unlicensed prescribing for paediatric patients in a public hospital, Indonesia.

### Method

A retrospective cross-sectional study of 200 randomly selected paediatric patients admitted to hospital between August and October 2014, collected patient details and all drugs prescribed. Licensed drugs were classified as off-label if there was a non-compliance with the Product Information for age, weight, indication, dose, frequency and route of administration, if there was a contraindication, special precautions or not recommended for children. Unlicensed drugs were those not approved for use in Indonesia. The main outcome was the prevalence of off-label or unlicensed prescribing to infants, children and adolescents and the impact of age group on off-label prescribing.

### Results

A total of 200 patients received 1961 medicines of which 1807/1961 (92.1%) were licensed and 154/1961 (7.9%) were unlicensed. There were 1403/1961 (71.5%) drugs prescribed off-label. More than half of the total drugs (n = 1066; 54.4%) were administered parenterally. Every patient was prescribed at least one off-label drug. Indication (n = 810; 34.6%) was the most common reason for off-label prescribing. Ranitidine was the most frequent drug prescribed off label. Darplex® (dihydroartemisinin and piperaquine), although manufactured in Indonesia, was unlicensed. There was a significant difference between age group and off-label prescribing in that children were prescribed significantly less off-label drugs (p<0.0003).

**Funding:** The authors received no specific funding for this work.

**Competing interests:** The authors have declared that no competing interests exist.

## Conclusion

This study revealed a high prevalence of off-label and unlicensed drug use in paediatric patients in this hospital, exposing them to drug treatments or regimens that had not been approved by regulatory authorities. The high incidence of invasive parenteral prescribing is of concern for paediatric patients. Incentives are needed to encourage specific drug evaluation in paediatric populations.

## Introduction

Many drugs prescribed for paediatric patients have undergone limited scientific evaluation with only an estimated one-third of them having been evaluated in clinical trials in this population.[1] Clinical trials that evaluate drug efficacy, safety and quality are required for registration.[2, 3] However, until recently clinical trials were limited in paediatric populations due to perceived ethical concerns regarding patient risk from research.[1, 4] Consequently, many drugs prescribed for children have been assumed to be safe and effective based on studies performed in adult populations.[5]

Children should not be perceived to be 'small adults' since their pharmacokinetics and pharmacodynamics vary throughout childhood, leaving large uncertainty about the relative efficacy of drugs with respect to adult populations.[6–8] Despite this, many drugs are used 'off-label' or outside their licensed recommendations. The use of off-label medicines refers to the use of a licensed drug at a different dose, indication, age or route of administration from that stated in the approved Product Information (PI).[2, 9] Unlicensed medicines include the reformulation of a licensed drug to provide a modified dosage form considered more suitable for administration but not licensed for paediatric administration, or the specific administration of drugs not licensed in Indonesia.[5]

Despite the introduction of several initiatives to stimulate more research into investigating drug use in children, including the Food and Drug Administration's (FDA's) 'Pharmaceuticals for Children Act' which was last amended in 2007, and the European Medicines Agency's (EMA's) 'European's Union's Paediatric Regulation' in 2007, it has been reported that children are still under-represented with respect to participating in clinical trials and more work is required to ensure safe use of medicines in children.[10] It is estimated that for the last decade, in hospital settings, the use of off-label drugs ranged from 12.2% to 70.6%.[3] Developing countries are most acutely affected[1] since persons aged between zero to 18 years make up a large proportion of the population[11] and are most vulnerable to diseases.[2]

The prescribing of off-label and unlicensed medicines may be unavoidable in situations where there is no alternative and the benefits potentially outweigh the risks.[1, 2] However, there is a potential public health risk for patients as there is uncertainty about efficacy and toxicity, that may result from off-label or unlicensed medicine use.[12] There may also be a risk associated with compounding drugs to produce a new formulation as there is often no safety information on drug interactions, stability or efficacy due to the potential changes in bioavailability.[2]

Indonesia, a developing country, requires drugs to be licensed by the National Agency of Drug and Food Control before they are marketed. In order to be granted a licensing number, a drug has to be evaluated to ensure it meets the requirements of safety, efficacy and quality. Drug licensing has two stages—pre-licensing and licensing.[13] Drugs prescribed or dispensed

off-label are not associated with a statutory responsibility of the prescribing physician or dispensing pharmacist.

The primary aims of this study were to determine the prevalence of off-label and unlicensed prescribing in paediatric patients in a medical ward of a public general hospital in Indonesia and to identify drugs most commonly prescribed off-label and unlicensed.

## Materials and methods

### Ethical approval

This study was approved by the human research ethics committee at Curtin University (approval number: RHDS-08-15) and by the Director of Abepura's Hospital, Papua, Indonesia (445/82.7/RSUD-Abe/ll/2015). Individual patient consent was not required as de-identified data were collected retrospectively.

### Data collection

This cross-sectional study collected medical record data retrospectively from a three-month period 1st August to 31st October 2014 in a paediatric medical ward in Abepura Hospital, Papua, Indonesia, which is a general tertiary hospital. No surgery patients were admitted to the medical ward and all patients were aged from 1 month to 14 years of age.

Of the total 307 patients admitted to the ward over this period, 200 were randomly selected using an on-line randomiser. This number was considered to give a reasonably accurate (±7%) estimate of the prevalence of off-label and unlicensed medicines in this ward. A data collection form was developed and used to collect data from patient medical records. Collected data were transferred to an Excel spread sheet which included information about patient demographics (age, date of birth, weight, gender, length of stay), reasons for admission, past medical history and prescribed drugs (name, indication, strength, dose, frequency, dosage form, route of administration and date of prescription). The following prescribing data were excluded from this study: traditional herbal medicines, standard intravenous replacement solutions, oxygen, total parenteral nutrition and blood products.

### Classification

Patients were classified according to the European Medicines Agency (EMA) age classification [14] as newborn infants (0 to 27 days), infants (28 day to 23 months), children (2 to 11 years) and adolescents (12 to 18 years). All drugs were coded using the World Health Organization (WHO) anatomical therapeutic chemical (ATC) classification.[15]

The licensing status of all medicines was determined using the database from the Indonesian National Agency of Drug and Food Control.[16] Drugs with no information on the database were classified unlicensed. Each licensed drug was checked against the Product Information (PI) of the Indonesian MIMS[17, 18] to determine whether or not it was off-label. If there was no information on MIMS, the Indonesian National Medicine Information[19] was consulted. Drugs were classified off-label for the following reasons:

1. Age/ Weight: Drugs prescribed outside the age or weight range from that in the PI. When there was no information for paediatric use, the drug was also considered off-label for age

2. Indication: The use of the drug was not for the indication(s) listed in the PI

3. Dose/ Frequency: Drugs prescribed at a lower or higher dose/ frequency from that listed in the PI. An allowance of ± 10% was permitted for rounding.

4. Route of administration: Drugs administered via a different route from that stated in the PI

5. Contraindication/not recommended/special precaution: Drugs where the PI stated they were contraindicated, drugs with unmet special precautions of specific age groups or those not recommended in the PI.

It was possible for a drug to be classified as off-label for more than one reason. If a drug was licensed but prescribed in a formulation that was unlicensed, then it was classified as unlicensed.

## Statistical analysis

Descriptive statistics were used for demographic data and to determine the extent of the frequency of off-label and unlicensed prescribing with respect to patients and with respect to the number of medicines prescribed. A Generalized Estimating Equation (GEE) was used to explore factors associated with a drug being given 'off-label' as opposed to 'on-label' (excluding the relatively small proportion of unlicensed drugs). This model was used because it took into account any correlations between the drugs being prescribed to the same patient. The following independent variables were included in the model: gender, age group, body weight, length of hospital stay, ATC drug category and diagnoses. Initially, all independent variables were included in the model, then the least significant was dropped, one at a time, until all variables remaining in the model were significantly associated with the outcome (a 'backwards elimination' strategy). Finally, pairwise interactions between independent variables remaining in the model were tested for statistical significance. All data were analysed using the SAS version 9.2 software, and a p-value $<0.05$ was taken to indicate a statistically significant association in all tests.

A Pearson's chi-square test was performed to compare the difference in off-label and unlicensed prescribing between the age groups. Where the numbers were too small for Chi-square to be considered accurate (expected frequency in at least one cell of the table less than 5), Fisher's Exact test was used instead.

## Results

### Patient demographics

Of 200 randomly selected medical records, 124 (62.0%) were males and 76 (38.0%) were females. The median age of patients was 1.7 years (range: one month– 13.2 years). The median weight of patients was 9.5 kg (range: 3.8–44.0 kg). The median length of hospital stay was 9 days (range: 2 to 33 days).

With respect to the age of paediatric patients, there were no newborns, 114 (57.0%) infants 80 (40.0%) children and 6 (3.0%) adolescents.

The 200 patients were diagnosed with 430 different indications. Acute gastroenteritis and/or diarrhoea was a common reason for patient admission and was listed as a diagnosis for 106/200 (53.0%) of admissions. Other reasons included dehydration (n = 93/200; 46.5%) and malaria (either *Plasmodium falciparum* or *Plasmodium vivax*) (n = 45/200; 22.5%).

### Drugs classified as off-label and unlicensed

A total of 1961 drugs were prescribed to 200 patients, of which 1807 (92.1%) were licensed. There were 154 (7.8%) unlicensed drugs and 1403 (71.5%) were prescribed off-label. More than half of the total drugs prescribed (n = 1066; 54.4%) were administered parenterally.

**Table 1. The 10 most commonly prescribed drugs with their off-label and unlicensed status.** The numbers in the columns are the number of prescriptions of the drug and the percentage of these which are on/off-label or unlicensed.

| Ten most common medications of 1961 prescribed drugs | Prescribed On-label | | Prescribed Off-label | | Unlicensed medicines | |
|---|---|---|---|---|---|---|
| | N | (%) | n | (%) | n | (%) |
| Paracetamol (n = 246; 12.5%) | 59 | 24.0 | 187 | 76.0 | 0 | 0.0 |
| Ranitidine (n = 230; 11.7%) | 1 | 0.4 | 229 | 99.6 | 0 | 0.0 |
| Cefotaxime (n = 187; 9.5%) | 26 | 13.9 | 161 | 86.1 | 0 | 0.0 |
| Ondansetron (n = 174; 8.9%) | 0 | 0.0 | 174 | 100.0 | 0 | 0.0 |
| Zinc sulfate (n = 134; 6.8%) | 97 | 72.4 | 37 | 27.6 | 0 | 0.0 |
| Liprolac®* (n = 119; 6.1%) | 88 | 74.0 | 31 | 26.1 | 0 | 0.0 |
| Cefixime (n = 115; 5.9%) | 12 | 10.4 | 88 | 76.5 | 15 | 13.0 |
| Gentamicin (n = 104; 5.3%) | 9 | 8.7 | 95 | 91.4 | 0 | 0.0 |
| Ceftriaxone (n = 69; 3.5%) | 16 | 23.2 | 53 | 76.8 | 0 | 0.0 |
| Artesunate (n = 57; 2.9%) | 0 | 0.0 | 57 | 100.0 | 0 | 0.0 |

*Liprolac® (2.5g powder) sachets consist of viable cells 1,25 x 109 CFU (*Streptococcus thermophilus* 10 mg, *Lactobacillus rhamnosus* 3 mg, *Lactobacillus acidophilus* 3 mg, *Bifidobacterium longum* 1.25 mg, *Bifidobacterium bifidum* 1.25 mg), polydextrose 869.63 mg, fructooligosaccharide 375 mg, lactulose mixed powder 125 mg, vitamin C 35 mg, vitamin E 8.125 mg, vitamin A 3.60 mg, pyridoxine HCl 1.13 mg, vitamin B2 0.75 mg, thiamine HCl 0.70 mg).

The ten most commonly prescribed drugs overall, including those that were prescribed off-label or unlicensed are shown in Table 1 (the complete list of all drugs is available as S1 Table). Of the 1403 off-label medicines, ranitidine, ondansetron, gentamicin and artesunate showed high levels of this prescribing. Table 2 shows the main reasons, and particular combinations of reasons, for off-label use of the medicines listed in Table 1 (an alphabetical list of all drugs prescribed off-label is included as a table in S2 Table). Several drugs including ranitidine and ondansetron were off-label for several reasons. Both age/weight and indication were high contributors to off-label prescribing. Cefotaxime and ceftriaxone were frequently prescribed in the absence of any indication for their need. This often related to them being prescribed for gastroenteritis. No microbiology data were collected to ascertain any bacterial involvement.

Based on the total number of 154 unlicensed medicines, the ten drugs most frequently prescribed were Darplex® (dihydroartemisinin and piperaquine) (n = 30; 19.5%), cough powder '14' (a combination of ambroxol, guaifenesin, chlorpheniramine maleate and vitamin C) (n = 28; 18.2%), primaquine (n = 18; 11.7%), cefixime (n = 15; 9.7%), elemental iron (n = 6; 3.9%), phenobarbitone (n = 6; 3.9%), cough powder '3' (a combination of ambroxol, dexamethasone, salbutamol and tremenza) (n = 5; 3.2%), sucralfate (n = 5; 3.2%), colistin sulphate (n = 4; 2.6%) and fluconazole (n = 4; 2.6%). Among unlicensed medicines, Darplex® although manufactured in Indonesia was not a registered product in Indonesia. Both sucralfate and elemental iron were imported. All other drugs were compounded preparations.

## Drug use based on the anatomical therapeutic chemical (ATC) classification

The most commonly prescribed ATC category was the alimentary tract and metabolism (Class A) (n = 776/1961; 39.6% of all drugs), with ranitidine (n = 230; 29.6% of the 776 drugs within this class) and ondansetron (n = 174/776; 22.4%) most frequently prescribed. Ondansetron was always prescribed off-label and ranitidine 99.6% of the time. The second most frequently prescribed ATC category was anti-infectives (Class J) (n = 533/1961; 27.2%) with cefotaxime, cefixime and gentamicin most commonly prescribed. Usually gentamicin was off-label 95/104

**Table 2. The reasons, or particular combination of reasons, for off-label use of the ten most common medicines shown in Table 1.** Under each drug name are the number of prescriptions (n), and the number of patients (N).

| Drug | Reason for off-label classification | | | | | |
|---|---|---|---|---|---|---|
| | Frequency (n) | Age/ weight | Indication | Dose/ frequency | Contra-indication | Special Precautions |
| Ranitidine (n = 229 N = 196) | 219 | ■ | ■ | | | ■ |
| | 6 | ■ | | | | ■ |
| | 1 | ■ | | | | |
| | 1 | ■ | ■ | | | |
| | 1 | ■ | | ■ | | |
| | 1 | ■ | | | ■ | |
| Paracetamol (n = 187, N = 138) | 148 | ■ | | | | |
| | 39 | | | ■ | | |
| Ondansetron (n = 174, N = 135) | 127 | ■ | ■ | | | |
| | 45 | | ■ | | | |
| | 1 | ■ | | | | ■ |
| | 1 | | | ■ | | ■ |
| Cefotaxime (n = 161, N = 132) | 156 | | ■ | | | |
| | 4 | | ■ | | ■ | |
| | 1 | | ■ | | | |
| Gentamicin (n = 95, N = 85) | 56 | | ■ | ■ | | |
| | 23 | | ■ | | | |
| | 16 | | | ■ | | |
| Cefixime (n = 88, N = 88) | 41 | | ■ | | | |
| | 27 | | ■ | ■ | | |
| | 9 | | | ■ | | |
| | 6 | | ■ | | ■ | |
| | 5 | | | | ■ | |
| Artesunate (n = 57, N = 46) | 38 | ■ | | | | |
| | 17 | ■ | | ■ | | |
| | 1 | | | ■ | | |
| | 1 | ■ | ■ | | | |
| Ceftriaxone (n = 53, N = 48) | 51 | | ■ | | | |
| | 3 | | ■ | ■ | | |
| Zinc sulfate (n = 37, N = 36) | 35 | | | ■ | | |
| | 2 | ■ | | | | |
| Liprolac (n = 31, N = 30) | 31 | | | ■ | | |

The interpretation of results in Table 2 is as follows—for 23 prescriptions, gentamicin was off-label due to indication. However, for 56 (separate) prescriptions, gentamicin was off-label due to indication as well as dose/ frequency. In total there were 95 off-label prescriptions for gentamicin, for a total of 85 patients.

(91.3%). The third most commonly prescribed ATC category was the nervous system (Class N) (n = 274/1961; 14.0%).

Amongst the unlicensed drugs, the most common ATC category was antiparasitic products, insecticides and repellents (Class P) (n = 49, 31.8% of 154 unlicensed drugs) which included Darplex® (n = 29; 59.2% of the 49 drugs in the class) and primaquine (n = 18/49; 36.7%).

## Patient prescribing

The median number of drugs prescribed per person was 9 drugs (range: 4–20). All 200 patients (100%) were given at least one off-label drug and 108 (54%) of them received at least one

**Table 3. Level of off-label and unlicensed prescribing to patients according to age classification.**

| Age groups | Patients with at least one prescribed drug that was: | | | |
| --- | --- | --- | --- | --- |
| | Off-label | | Unlicensed | |
| | n = 200 | (%) | n = 108 | (%) |
| **Infants** (n = 114; 57.0%) | 114 | 100 | 60 | 52.6 |
| **Children** (n = 80; 40.0%) | 80 | 100 | 43 | 53.8 |
| **Adolescents** (n = 6; 3.0%) | 6 | 100 | 5 | 83.3 |

unlicensed medicine (Table 3). There appeared to be no difference between age groups in the proportion of patients who received at least one unlicensed drug (Fisher's Exact test; p = 0.38). The distribution of off-label and unlicensed prescribing of the 1961 drugs in infants, children and adolescents is summarized in Table 4. While it appeared that the use of off-label drugs was more common for adolescents than infants, no significant association was evident between age-group and the use of off-label drugs (GEE model, univariate p = 0.090).

## Reasons for off-label and unlicensed prescribing

Of the 1403 drugs which were prescribed 'off-label', the most common reason given was indication (diagnosis) (n = 810; 57.7%), followed by age/weight (n = 655; 46.7%), dose/frequency (n = 579; 41.3%) and the use of contraindicated drugs or those with special precautions or not recommended to be used in the patient age prescribed (n = 294; 21.0%). No drug was off-label for route of administration. Neither length of stay (2–3 days vs 4–6 days, p = 0.063; 7+ days vs 4–6 days, p = 0.11) nor gender (p = 0.92) were associated with the rate of off-label prescribing.

The GEE model shows several factors that were independently associated with off-label prescribing (Table 5). While age group appeared to be significantly associated with off-label prescribing based on univariate analysis (Infants: 77.7%; Children: 76.4%; Adolescents: 91.2%; with the p-value for Adolescents vs others p = 0.01), this difference did not persist after adjustment for the other independent variables in the model. As there were no neonates admitted to this ward, it was not possible to determine if off-label prescribing would have been more or less frequent in this age group.

Other independent variables in the GEE model were dropped as they appeared to be not significantly associated with the outcome (including gender, body weight, length of hospital stay, ATC drug category and diagnoses). The p-values for the variables which were dropped were: gender (p = 0.92), weight (0.20). We divided LOS into short (2–3 days), medium (4–6 days), and long (7+ days). With medium stay as the reference, the p-value for short was p = 0.063, and for long stay it was p = 0.11, and therefore it was also dropped from the final model. ATC categories and diagnoses which were not included in the final model were dropped because they showed no significant association. It is notable that an indication of tuberculosis was associated with lower off-label prescribing (p = 0.0037), and malaria was associated with a higher rate (p = 0.0004). The specific ATC codes also influenced off-label

**Table 4. Frequency of off-label and unlicensed prescribing based on total drugs prescribed.**

| Age groups | Medicines not off-label or unlicensed | | Off-label medicines | | Unlicensed medicines | |
| --- | --- | --- | --- | --- | --- | --- |
| | n = 404 | (%) | n = 1403 | (%) | n = 154 | (%) |
| **Infants (n = 1144)** | 236 | 20.6 | 824 | 72.0 | 84 | 7.3 |
| **Children (n = 755)** | 163 | 21.6 | 527 | 69.8 | 65 | 8.6 |
| **Adolescents (n = 62)** | 5 | 8.1 | 52 | 83.9 | 5 | 8.1 |

**Table 5. Analysis of association with a drug being defined as off-label using a Generalized Estimating Equation (GEE).** (n = 1807 licensed drugs prescribed on or off-label). The "n" in the first column shows the total number of drugs prescribed, while the "n" in the second column shows the number which were off-label.

| Variable | Number of off-label drugs n (%) | Odds Ratio | 95% Confidence Interval | p-value |
|---|---|---|---|---|
| **Tuberculosis** | | | | |
| No (n = 1677) | 1316 (78.5) | 1 (reference) | | |
| Yes (n = 130) | 87 (66.9) | 0.53 | 0.35–0.82 | 0.0037 |
| **Malaria** | | | | |
| No (n = 1435) | 1087 (75.8) | 1 (reference) | | |
| Yes (n = 372) | 316 (85.0) | 1.73 | 1.28–2.34 | 0.0004 |
| **ATC Class** | | | | |
| N* class (n = 266) | 200 (75.2) | 1 (reference) | | |
| A# class (n = 764) | 529 (69.2) | 0.68 | 0.49–0.94 | 0.0205 |
| Other classes (n = 237) | 208 (87.8) | 1.92 | 1.16–3.20 | 0.0116 |
| **ATC Class J** : Bronchitis** | | | | |
| No: No (n = 1178) | 868 (73.7) | 1 (reference) | | |
| No: Yes (n = 123) | 102 (82.9) | 1.81 | 1.08–3.02 | 0.0231 |
| Yes: No (n = 447) | 412 (92.2) | 3.87 | 2.18–6.89 | <0.0001 |
| Yes: Yes (n = 59) | 21 (35.6) | 0.20 | 0.11–0.35 | <0.0001 |

* = drugs affecting the nervous system;

# = alimentary and metabolism drugs;

** = anti-infectives.

prescribing, with Class A (alimentary tract drugs) having a lower rate of off-label use than Class N (nervous system drugs), and other classes having a higher rate of off-label use. ATC Class J (anti-infective agents) showed an interaction with bronchitis so that in the absence of bronchitis, Class J drugs were more likely to be used off-label, but when bronchitis was present, Class J drugs were less likely to be used off-label. The number of cases where both bronchitis was present and a Class J drug was used, was small (n = 59), but the difference in use in an off-label fashion was quite clear, and therefore statistically significant.

## Discussion

To the best of our knowledge this is the first study to report the prevalence of off-label and unlicensed prescribing in a paediatric medical ward in a public general tertiary hospital setting in Indonesia. Previous studies in Indonesia have reported on the prevalence of off-label and unlicensed prescribing in 67 paediatric inpatients with nephrotic syndrome (aged less than 18 years)[20] and off-label use of anticonvulsants at a private hospital (age not specified).[21] It is not possible to make a direct comparison between the results of these studies with the current findings due to the differences in setting and age of the paediatric patients.

The current study found that off-label and unlicensed prescribing was a common practice in this Indonesian hospital setting. Similar findings have been reported in other studies.[11, 22–28] The median number of drugs prescribed was relatively high at nine per patient (range: 4–20 drugs). In a hospital study in Malaysia (also a Southeast Asian country) in which the median age of the study population was two years, researchers reported that the median number of drugs prescribed per child was four (range 1–52).[28] In our study population, the median age was similar (1.7 years) but the median number of drugs prescribed was much higher. Similar findings were also reported in a study in Finland, in which the median number

of drugs in preterm infants and children aged 2–11 years, was 15 (range 2–29) and 9 (range 0–36) respectively.[25] In the current study, there was also a high rate of injection prescribing. Injection rates are often high in developing countries because of a perception that this route of administration is more effective.[29] The high rate of injection prescribing is also of concern to the World Health Organization.[30] These findings raise potential public health issues.

## The extent of off-label medicines

A high prevalence of off-label prescribing was found (71.6%) compared to other studies in similar settings in Palestine (31.3%),[11] Germany (31%),[24] Australia (31.8%),[22] Slovak Republic (22%),[31] and Finland (42%).[25] Although these differences may be partly due to the different patient populations with different indications compared to other countries, they may also be due to the use of different definitions of off-label.[12] Unlike some previous studies, this study included as off-label drugs, those which were contraindicated, those with special precautions, and those drugs not recommended for use in children. Moreover, there may be differences in the contents of the Product Information (PI) for the same active ingredient between countries due to differences in registration data submitted.[12, 32]

## Off-label drugs

Ranitidine, which was the second most frequently prescribed drug overall, was the most commonly prescribed off-label drug. It was off-label for age and indication because there was no specific age in children at which ranitidine was licensed for parenteral use and it was also prescribed without a clear diagnosis (indication). Several other studies have reported the use of ranitidine among the ten most commonly prescribed off-label drugs. However, these studies were conducted in a surgical ward, PICU, neonate intensive care unit, and paediatric emergency unit.[26, 33] All drugs in the 'cardiovascular system' classifications were prescribed off-label due to dose/ frequency and indication.

## Reasons for a drug to be off-label

In a recent review, researchers reported that the hospital setting influenced the reason for off-label prescribing, with age and dose most commonly reported reasons for off-label prescribing in neonatal intensive care units; dose, age and indication in paediatric intensive care units; dose ad age in general paediatric wards and dose in the emergency department.[12] In the current study, the most common reason for off-label prescribing was for indication. For example, a parenteral formulation of ondansetron was only indicated for nausea in children with chemotherapy but none of the children on this ward was undergoing chemotherapy hence ondansetron was off-label for all the prescriptions. Other studies have also reported that ondansetron was prescribed off-label due to indication.[34, 35] In addition, the prescribing pattern of antibiotics in this hospital may have contributed significantly to the off-label category based on the indication. For example, cefotaxime was prescribed for many acute gastrointestinal cases although there was no registered indication in the PI for acute gastroenteritis. Age category was associated with a small but statistically significant lower level of off-label prescribing for children (aged 2–11 years) compared to infants or adolescents. The rate of off-label prescribing was however high for all age groups. This is concerning as several studies have reported that off-label prescribing is a risk factor for developing adverse reactions.[36, 37] As highlighted by Bonati et al.,[38] when drugs are prescribed off-label for specific diseases for which indications are not included in the manufacturer's PI, evidence for their use should be established.[38] Additionally, such prescribing may raise medico-legal issues.[39]

It is notable that the prescribing of antibiotics (often as injections) in the management of gastroenteritis was high. No microbiology testing was performed on any potential pathogens, however antibiotics were not registered for this diagnosis. Where antibiotics were an appropriate selection, a lower level of off-label prescribing occurred. Inappropriate antibiotic prescribing has the potential to lead to resistance.[40]

## Unlicensed prescribing

The percentage of unlicensed medicines prescribed in this study (7.9%) conformed within the range of 5.5%–28% reported by other studies in similar settings.[11, 25] A number of studies in developing transitional countries such as Palestine[11] and Malaysia[28] also reported that most unlicensed drugs were extemporaneous preparations produced from capsules or tablets in order to obtain a lower dosage which was not available in the registered products. For instance, the preparation of 3 mg primaquine was made from a 15 mg tablet crushed to aliquot the desired dosage. Such formulations introduce uncertainty of the stability and bioavailability of the mixtures.[2]

The use of extemporaneous preparations was also aimed to reduce the number of oral drug administrations. Darplex® was the most frequent unlicensed drug because it was unregistered in Indonesia. It is a combination of dihydroartemisinin and piperaquine phosphate, which is one of the WHO recommended combination drugs to treat malaria. The reason why Darplex® was unregistered in Indonesia is unknown.

## Informed consent

There was no informed consent documented for the administration of off-label or unlicensed medicines in this hospital. Informed consent may be appropriate when drugs without the support of clinical data or high quality evidence are prescribed of use in children.[22]

## Limitations

Despite the high prevalence of off-label use in paediatric patients reported in this study, there were some limitations. As this was a retrospective study, some data may have been missing. Further, the data were collected in 2015, so it is possible some prescribing has changed since that date. Although the data were collected over a three month period from August to October, as the hospital was located at sea level near the equator, it has a tropical climate and little seasonal variation. Hence potential biased analysis toward seasonal effects is unlikely. As the study did not include neonates and adolescents older than 14 years of age, it may not be an accurate representation of the complete paediatric population. Also, this study may not be comparable to other parts of Indonesia as they may have different patterns of diseases. An evaluation of the safety and efficacy of the drugs prescribed off-label in this study was outside the scope of this research however, there is a need for these evaluations in future studies.

## Conclusion

This study reports a high prevalence of both off-label and unlicensed prescribing in paediatric patients in a general medical ward in a hospital setting in Indonesia. Antibiotic prescribing for unlicensed indications was a major contributor and potential public health issue related to these findings. Specific disease states such as tuberculosis resulted in significantly lower off-label prescribing. Prescribers in Indonesia should be made aware of the findings of this study.

## Supporting information

**S1 Table. All drugs with off-label and unlicensed status.**
(DOCX)

**S2 Table. Reasons, or combination of reasons, for off-label prescribing.**
(DOCX)

## Acknowledgments

Authors wish to acknowledge the assistance of the medical records department of the research hospital for their support and assistance.

## Author Contributions

**Conceptualization:** Bruce Sunderland, Richard Parsons, Petra Czarniak.

**Data curation:** Brechkerts Lieske Angruni Tukayo.

**Formal analysis:** Brechkerts Lieske Angruni Tukayo, Richard Parsons, Petra Czarniak.

**Investigation:** Brechkerts Lieske Angruni Tukayo.

**Methodology:** Bruce Sunderland, Petra Czarniak.

**Supervision:** Bruce Sunderland, Richard Parsons, Petra Czarniak.

**Validation:** Petra Czarniak.

**Writing – original draft:** Brechkerts Lieske Angruni Tukayo, Bruce Sunderland, Petra Czarniak.

**Writing – review & editing:** Brechkerts Lieske Angruni Tukayo, Bruce Sunderland, Richard Parsons, Petra Czarniak.

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
