## [Decision Letter · Decision Letter 0]

2 Sep 2019

PONE-D-19-20113

High prevalence of off-label and unlicensed paediatric prescribing in a hospital in Indonesia

PLOS ONE

Dear Dr Czarniak,

Thank you for submitting your manuscript to PLOS ONE. After careful consideration, we feel that it has merit but does not fully meet PLOS ONE’s publication criteria as it currently stands. Therefore, we invite you to submit a revised version of the manuscript that addresses the points raised during the review process.

We would appreciate receiving your revised manuscript by Oct 17 2019 11:59PM. To enhance the reproducibility of your results, we recommend that if applicable you deposit your laboratory protocols in protocols.io, where a protocol can be assigned its own identifier (DOI) such that it can be cited independently in the future. For instructions see: http://journals.plos.org/plosone/s/submission-guidelines#loc-laboratory-protocols

We look forward to receiving your revised manuscript.

Kind regards,

Jinn-Moon Yang

Academic Editor

PLOS ONE

Journal Requirements:

2. Please provide additional details regarding participant consent. In the ethics statement in the Methods and online submission information, please ensure that you have specified (1) whether consent was suitably informed and (2) what type you obtained (for instance, written or verbal). Since your study included minors under age 18, state whether you obtained consent from parents or guardians. If the need for consent was waived by the ethics committee, please include this information.

Additional Editor Comments (if provided):

Reviewers' comments:

Reviewer's Responses to Questions

**Comments to the Author**

1. Is the manuscript technically sound, and do the data support the conclusions?

Reviewer #1: Partly

Reviewer #2: No

Reviewer #3: Yes

2. Has the statistical analysis been performed appropriately and rigorously? 

Reviewer #1: No

Reviewer #2: I Don't Know

Reviewer #3: No

3. Have the authors made all data underlying the findings in their manuscript fully available?

Reviewer #1: No

Reviewer #2: No

Reviewer #3: No

4. Is the manuscript presented in an intelligible fashion and written in standard English?

Reviewer #1: Yes

Reviewer #2: No

Reviewer #3: Yes

5. Review Comments to the Author

Reviewer #1: High prevalence of off-label and unlicensed pediatric prescribing in a hospital in Indonesia

In this manuscript, the authors aimed to determine the prevalence of off-label and unlicensed prescribing for pediatric patients in a public hospital, Indonesia. The study is interesting, however, a number of issues should be addressed before its publication.

Major issues:

1. The authors used a GEE model to explore factors associated with a drug being given off-label as opposed to on-label. Several variables (gender, age group, body weight, length of hospital stay, ATC drug category and diagnoses) were included in the model. The results in Table 5 were unclear.

(1) There were no results about gender, body weight, length of hospital stay.

(2) The factors of GEE model in table 5 were not convinced. For example, unreasonable age grouping (infant + adolescent vs child), unknown reasons for ATC class (alimentary and metabolism drugs + anti-infectives + drugs affecting the nervous system vs others).

(3) Why was the variable "tuberculosis" in the model? Since the main indications were acute gastroenteritis, diarrhea, dehydration, and malaria in 430 indications, they should be considered and discussed in the manuscript.

(4) More information about factor "diagnoses" should be addressed.

(5) Were there interaction terms in the GEE model?

2. In table 2, why the number of reasons for ranitidine were combined (age/weight + indication + special precautions - 219), but the number of reasons for paracetamol were separated (age/weight - 148; dose/frequency - 39)? The number of reasons for ondansetron, gentamicin, cefixime, and artesunate were unclear. Some reasons were repeated in the same drug. For example, gentamicin (indication; dose/frequency - 56; indication - 23).

3. The result of the chi-square test was doubtful (line 256, X^2(2) = 2.2 ). More information is required. Since the number of adolescents was few (n = 6), it should be aware of the fact that there might be a low expected value.

4. more information about p-value in table 4.

Minor issues:

1. Lines 150-159 were the reasons that drugs were classified as off-label. Number labeling needs to be modified.

2. Lines 180-181: when the authors did chi-square test, two or three age groups were compared?

3. Require more information about proportions. The author stated many proportions in the manuscript, and the total number changed with different themes. It would be clearer if the denominator of the ratio or the total number is written.

Reviewer #2: The present work provide a descriptive analysis of the prevalence of off-label and unlicensed pediatric prescription in a hospital in Indonesia. The analysis is based on 200 randomly selected patients whose hospital admission were within Aug to Oct 2014. The prevalence and most commonly drugs with reasons for off-label use were reported. Issues of this work are listed below.

1. The size of study population is very small with only 200 patients. Why they consider only 200 randomly selected patients rather than include all patients?

2. The p-value for age group comparison should be further checked given vey similar percentages of the two groups.

3. As they mentioned in the section of Limitations, the data was collected five years ago that may not reflect the actual prevalence of off-label and unlicensed prescriptions. Also, a very short period from Aug to Oct was considered that result in a potentially biased analysis toward seasonal effects. The prevalence calculated in this study is therefore not representative for the status of a hospital in Indonesia. They may consider to enlarge the size of study population and cover at least one year to produce meaningful results.

4. A few grammar errors and typos were found that should be corrected.

5. Instead of showing only the top 10 drugs, they are sugguested to provide a detailed data sheet as supplementory data.

Reviewer #3: In this manuscript, the authors presented the statists of the off-label and unlicensed paediatric prescribing in Indonesia hospital. Data of 200 patients received 1961 medicines were collection and 7.9% and 71.5% of medicines were unlicensed and off-label used, respectively. Especially, the authors indicated that anitidine was the most frequent drug prescribed off label. The topic is interesting but there are some issues about the manuscript which need to be dealt with before accepted:

1. The data was collected only from a 3-month period in a hospital. Is it possible the statists have the seasonality and regionality?

2. The reason which using randomly selected 200 instead of 307 patients is unclear.

3. It would be interesting to have the multi-factor ANOVA or other statistical test using indications and drugs classification.

4. Ranitidine was the most frequent drug prescribed off label. The relationship between Ranitidine and indications should be discussed.

6. PLOS authors have the option to publish the peer review history of their article (what does this mean?). If published, this will include your full peer review and any attached files.

Reviewer #1: No

Reviewer #2: No

Reviewer #3: No

---

## [Author Response · Author response to Decision Letter 0]

17 Oct 2019

• A rebuttal letter that responds to each point raised by the academic editor and reviewer(s). This letter should be uploaded as separate file and labeled 'Response to Reviewers'. 

This has been provided.

• A marked-up copy of your manuscript that highlights changes made to the original version. This file should be uploaded as separate file and labeled 'Revised Manuscript with Track Changes'.

This has been provided.

• An unmarked version of your revised paper without tracked changes. This file should be uploaded as separate file and labeled 'Manuscript'.

This has been provided.

Journal Requirements:

We have checked this and consider our manuscript meets the requirements.

2. Please provide additional details regarding participant consent. In the ethics statement in the Methods and online submission information, please ensure that you have specified (1) whether consent was suitably informed and (2) what type you obtained (for instance, written or verbal). Since your study included minors under age 18, state whether you obtained consent from parents or guardians. If the need for consent was waived by the ethics committee, please include this information.

We have added the following statement to the manuscript: ‘Individual patient consent was not required as de-identified data were collected retrospectively’

5. Review Comments to the Author

Reviewer #1: High prevalence of off-label and unlicensed pediatric prescribing in a hospital in Indonesia

In this manuscript, the authors aimed to determine the prevalence of off-label and unlicensed prescribing for pediatric patients in a public hospital, Indonesia. The study is interesting, however, a number of issues should be addressed before its publication.

Major issues:

1. The authors used a GEE model to explore factors associated with a drug being given off-label as opposed to on-label. Several variables (gender, age group, body weight, length of hospital stay, ATC drug category and diagnoses) were included in the model. The results in Table 5 were unclear.

(1) There were no results about gender, body weight, length of hospital stay.

As described in the Methods section, we included all the independent variables named (including gender, age group, body weight, length of hospital stay, ATC drug category and diagnoses) in the GEE model, but then we dropped variables which appeared to be not significantly associated with the outcome. The p-values for the variables which were dropped were: gender (p=0.92), age group (p=0.09), weight (0.20). We divided LOS into short (2-3 days), medium (4-6 days), and long (7+ days). With medium stay as the reference, the p-value for short was p=0.063, and for long stay it was p=0.11, and therefore it was also dropped from the final model. ATC categories and diagnoses which were not included in the final model were dropped because they showed no significant association. This has been included in the revised manuscript.

(2) The factors of GEE model in table 5 were not convinced. For example, unreasonable age grouping (infant + adolescent vs child), unknown reasons for ATC class (alimentary and metabolism drugs + anti-infectives + drugs affecting the nervous system vs others).

Age-group was classified according to the European Medicines Agency, as described in the Methods section. The reason why the Infants and adolescents were initially grouped is that, compared to Infants, the odds ratio associated with adolescents was not statistically significantly different, and therefore the term for adolescents was dropped from the model. We have modified the model, reviewing all possible pairwise interactions, and now we find that the age group does not appear to significantly influence the model, as mentioned above. When including terms for the ATC level, we found that ATC groups A, J and N accounted for 86.6% of the drugs (43%, 28.5% and 15% respectively) with all other classes accounting for 13.4%. No single class (other than A, J or N) accounted for more than 4.5% of the drugs, and therefore this set of drugs were classified as ‘Other ATC group’. 

(3) Why was the variable "tuberculosis" in the model? Since the main indications were acute gastroenteritis, diarrhea, dehydration, and malaria in 430 indications, they should be considered and discussed in the manuscript.

We have subsequently revisited the final model, and found that the interaction between ATC class A and bronchitis was the dominant feature that then led to a simpler interpretation of the results. Tuberculosis is still included in the model because the odds of receiving an off-label drug was lower for those with this condition than those without. This has been modified in the manuscript.

(4) More information about factor "diagnoses" should be addressed.

We used the diagnoses which were recorded in patient medical records, which were transferred to the data sheet (gastro, dehydration, malaria, vomiting, obs febris, bronchitis, respiratory tract infection, tuberculosis, and others. The ‘other’ group included n=79 (18.4%) of diagnoses.

(5) Were there interaction terms in the GEE model?

We have included a statement in the Methods that we examined all pairwise interactions for all the main effects which appeared significantly associated with the outcome. We have revisited the final model. We had previously included pairwise interactions, but after a thorough re-analysis, we found that the interaction between ATC class A and bronchitis was the dominant term. Other interactions appeared not to add further to the model. Table 5 has been updated with the new analysis.

2. In table 2, why the number of reasons for ranitidine were combined (age/weight + indication + special precautions - 219), but the number of reasons for paracetamol were separated (age/weight - 148; dose/frequency - 39)? The number of reasons for ondansetron, gentamicin, cefixime, and artesunate were unclear. Some reasons were repeated in the same drug. For example, gentamicin (indication; dose/frequency - 56; indication - 23).

The table shows drugs that were more commonly prescribed off-label for a single reason or for several reasons. For example, for 23 prescriptions, gentamicin was off-label due to indication. However, for 56 (separate) prescriptions, gentamicin was off-label due to indication as well as dose/ frequency.

Table 2 has been modified for clarity and the above example has been included

3. The result of the chi-square test was doubtful (line 256, X^2(2) = 2.2 ). More information is required. Since the number of adolescents was few (n = 6), it should be aware of the fact that there might be a low expected value.

With the small number of cases in this table from the adolescents, it was found more appropriate to quote the p-value obtained from Fisher’s exact test. This has been updated in the text.

4. more information about p-value in table 4.

A sentence has been added just before the table to explain where this p-value came from (a univariate GEE model).

Minor issues:

1. Lines 150-159 were the reasons that drugs were classified as off-label. Number labeling needs to be modified.

Thankyou. This has been corrected.

2. Lines 180-181: when the authors did chi-square test, two or three age groups were compared?

In Tables 3 and 4, the p-values compare the 3 age groups for the rate of patients or drugs which were off-label vs on-label (unlicensed drugs excluded).

3. Require more information about proportions. The author stated many proportions in the manuscript, and the total number changed with different themes. It would be clearer if the denominator of the ratio or the total number is written.

We have reviewed the statements of percentages, and written explicitly what the denominator is (either patients or drugs). Thank you for pointing out this

Reviewer #2: The present work provide a descriptive analysis of the prevalence of off-label and unlicensed pediatric prescription in a hospital in Indonesia. The analysis is based on 200 randomly selected patients whose hospital admission were within Aug to Oct 2014. The prevalence and most commonly drugs with reasons for off-label use were reported. Issues of this work are listed below.

1. The size of study population is very small with only 200 patients. Why they consider only 200 randomly selected patients rather than include all patients?

A sample of 200 patients was selected for this study as it provides a sufficient sample size for regression and related analysis to be performed. It is evident from this study that we have identified a large number of individual data as off-label and unlicensed prescribing.

Examples of other studies that have included similar numbers of patients include:

• Landwehr C, Richardson J, Bint L, Parsons R, Sunderland B, Czarniak P. Cross-sectional survey of off-label and unlicensed prescribing for patients at a paediatric teaching hospital in Western Australia. PlosOne. 2019; 14(1):e0210237. Doi:10.1371/journal.pone.0210237. eCollection 2019

o The study involved 190 inpatient medication chart records and 1160 prescribed drugs

• Dornalles AD, Calegari LH, de Souza L, Ebone P, Tonelli TS, Carvalho CG. The unlicensed and off-label prescription of medications in general paediatric ward: an observational study. Curr Pediatr Rev. 2019; 15(1): 62-66

o This study involved 157 patients and 1328 prescriptions.

• Tefera YG, Gebresillassis BM, Mekuria AB, Erku DA, Seid N, Beshir HB. Off-label drug use in hospitalised children: a prospective observational study at Gondar University Referral Hospital, Northwestern Ethiopia. Pharmacol Res Perspect. 2017; 5(2):e00304

o The study involved 243 patients and 800 prescribed drugs

• Joret-Descout P, Prot-LabartheS, Brion F, Bataille J, Hartman JF, Bourdon O. Off-label and unlicensed utilisation of medicines in a French paediatric hospital. Int J Clin Pharm. 2015; 37(6): 1222-7

o The study involved 120 patients and a total of 315 prescription medicines 

2. The p-value for age group comparison should be further checked given vey similar percentages of the two groups.

In Tables 3 and 4, the p-values compare the 3 age groups for the rate of patients or drugs which were off-label vs on-label (unlicensed drugs excluded).

3. As they mentioned in the section of Limitations, the data was collected five years ago that may not reflect the actual prevalence of off-label and unlicensed prescriptions. Also, a very short period from Aug to Oct was considered that result in a potentially biased analysis toward seasonal effects. The prevalence calculated in this study is therefore not representative for the status of a hospital in Indonesia. They may consider to enlarge the size of study population and cover at least one year to produce meaningful results.

It was a random sample of data taken over the period of August to October so it would reflect prescribing for that period. However, even though the data were collected over a 3-month period, as the hospital was located at sea level near the equator, it has a tropical climate and little seasonal variation. Hence biased analysis toward seasonal effects are not likely.

We have replaced the following sentence in the section under ‘limitation’:

‘The time frame of this study may not account for any seasonal variation effect on the prescriptions pattern’

with:

‘Although the data were collected over a three month period from August to October, as the hospital was located at sea level near the equator, it has a tropical climate and little seasonal variation. Hence potential biased analysis toward seasonal effects is unlikely.’

This study was part of a Master of Clinical Pharmacy research project. It is therefore not possible to enlarge the size of the study population as the project has been completed.

4. A few grammar errors and typos were found that should be corrected.

Thank you. These have been corrected.

5. Instead of showing only the top 10 drugs, they are sugguested to provide a detailed data sheet as supplementory data.

We have included a supplementary data sheet showing the reasons for prescribing drugs off-label (Supplement 1 - All drugs with their off-label and unlicensed status).

Reviewer #3: In this manuscript, the authors presented the statists of the off-label and unlicensed paediatric prescribing in Indonesia hospital. Data of 200 patients received 1961 medicines were collection and 7.9% and 71.5% of medicines were unlicensed and off-label used, respectively. Especially, the authors indicated that anitidine was the most frequent drug prescribed off label. The topic is interesting but there are some issues about the manuscript which need to be dealt with before accepted:

1. The data was collected only from a 3-month period in a hospital. Is it possible the statists have the seasonality and regionality?

Although the data were collected over a three month period from August to October, as the hospital was located at sea level near the equator, it has a tropical climate and little seasonal variation. Hence potential biased analysis toward seasonal effects is unlikely.

Regionality was addressed in the limitations by stating: ‘This study may not be comparable to other parts of Indonesia as they may have different patterns of diseases.’

2. The reason which using randomly selected 200 instead of 307 patients is unclear.

A sample of 200 patients was selected for this study as it provides a sufficient sample size for regression and related analysis to be performed. It is evident from this study that we have identified a large number of individual data as off-label and unlicensed prescribing.

Examples of other studies that have included similar numbers of patients include:

• Landwehr C, Richardson J, Bint L, Parsons R, Sunderland B, Czarniak P. Cross-sectional survey of off-label and unlicensed prescribing for patients at a paediatric teaching hospital in Western Australia. PlosOne. 2019; 14(1):e0210237. Doi:10.1371/journal.pone.0210237. eCollection 2019

o The study involved 190 inpatient medication chart records and 1160 prescribed drugs

• Dornalles AD, Calegari LH, de Souza L, Ebone P, Tonelli TS, Carvalho CG. The unlicensed and off-label prescription of medications in general paediatric ward: an observational study. Curr Pediatr Rev. 2019; 15(1): 62-66

o This study involved 157 patients and 1328 prescriptions.

• Tefera YG, Gebresillassis BM, Mekuria AB, Erku DA, Seid N, Beshir HB. Off-label drug use in hospitalised children: a prospective observational study at Gondar University Referral Hospital, Northwestern Ethiopia. Pharmacol Res Perspect. 2017; 5(2):e00304

o The study involved 243 patients and 800 prescribed drugs

• Joret-Descout P, Prot-LabartheS, Brion F, Bataille J, Hartman JF, Bourdon O. Off-label and unlicensed utilisation of medicines in a French paediatric hospital. Int J Clin Pharm. 2015; 37(6): 1222-7

o The study involved 120 patients and a total of 315 prescription medicines

3. It would be interesting to have the multi-factor ANOVA or other statistical test using indications and drugs classification.

The ANOVA would be appropriate if the outcome is measured on a continuous scale. However, for the analyses presented, we have a binary outcome (off-label vs on-label). The ‘analagous’ analysis method for a multi-factor analysis of a binary outcome file with correlated measurements is the GEE which we have presented.

4. Ranitidine was the most frequent drug prescribed off label. The relationship between Ranitidine and indications should be discussed.

A table of reasons for the use of ranitidine has been included in the manuscript (Table 3). Note that the total number of reasons is 246, for the 229 administrations of ranitidine, which were off-label.

---

## [Decision Letter · Decision Letter 1]

12 Nov 2019

PONE-D-19-20113R1

High prevalence of off-label and unlicensed paediatric prescribing in a hospital in Indonesia

PLOS ONE

Dear Dr Czarniak,

Thank you for submitting your manuscript to PLOS ONE. After careful consideration, we feel that it has merit but does not fully meet PLOS ONE’s publication criteria as it currently stands. Therefore, we invite you to submit a revised version of the manuscript that addresses the points raised during the review process.

We would appreciate receiving your revised manuscript by Dec 27 2019 11:59PM. To enhance the reproducibility of your results, we recommend that if applicable you deposit your laboratory protocols in protocols.io, where a protocol can be assigned its own identifier (DOI) such that it can be cited independently in the future. For instructions see: http://journals.plos.org/plosone/s/submission-guidelines#loc-laboratory-protocols

We look forward to receiving your revised manuscript.

Kind regards,

Jinn-Moon Yang

Academic Editor

PLOS ONE

Reviewers' comments:

Reviewer's Responses to Questions

**Comments to the Author**

1. If the authors have adequately addressed your comments raised in a previous round of review and you feel that this manuscript is now acceptable for publication, you may indicate that here to bypass the “Comments to the Author” section, enter your conflict of interest statement in the “Confidential to Editor” section, and submit your "Accept" recommendation.

Reviewer #1: (No Response)

Reviewer #2: (No Response)

Reviewer #3: (No Response)

2. Is the manuscript technically sound, and do the data support the conclusions?

Reviewer #1: Partly

Reviewer #2: No

Reviewer #3: Partly

3. Has the statistical analysis been performed appropriately and rigorously? 

Reviewer #1: Yes

Reviewer #2: N/A

Reviewer #3: No

4. Have the authors made all data underlying the findings in their manuscript fully available?

Reviewer #1: Yes

Reviewer #2: Yes

Reviewer #3: No

5. Is the manuscript presented in an intelligible fashion and written in standard English?

Reviewer #1: (No Response)

Reviewer #2: Yes

Reviewer #3: Yes

6. Review Comments to the Author

Reviewer #1: 1. The authors state that with the small number of cases in this table from the adolescents, it was found more appropriate to quote the p-value obtained from Fisher's exact test. But in the section of statistical analysis of the revised manuscript, they didn't mention about Fisher's exact test.

2. Did the authors consider to exclude six adolescents from GEE model and Fisher's exact test / Chi-square test? Because there were only six adolescent patients in the data, the sample size of the adolescents was small. The situation of six adolescent patients could be discussed separately.

3. The items in Table 2 are confused. Some drugs have two frequencies (e.g., paracetamol, ondansetron, ...). The cumulative frequency of age/weight, indication, dose/frequency, and special precautions for ranitidine is 243, which was not equal to the frequency of ranitidine in Table 2. It might be that the frequency of ranitidine in Table 2 was counted as combination of reasons, but the frequency in Table 3 was not. Even so, it's easy to confuse. Information about Table 2 might be addressed in the context or the footnote, but not in the title of Table 2.

Reviewer #2: The manuscript has been largely improved by addressing most of the issues raised from previous review. But there is still one issue to be solved, the title does not fit the context. They are suggested to append 'during the period Aug.-Oct. 2014' to the title of manuscript to reflect the context and avoid the over-claiming issue.

Reviewer #3: The reason is still unknown why the size of study population is only 200 randomly selected patients rather than include all 307 patients? Although 200 is enough to provides a sufficient sample size for regression and related analysis to be performed, how about all 307 patients? Please show the results of non selected dataset.

7. PLOS authors have the option to publish the peer review history of their article (what does this mean?). If published, this will include your full peer review and any attached files.

Reviewer #1: No

Reviewer #2: No

Reviewer #3: No

---

## [Author Response · Author response to Decision Letter 1]

4 Dec 2019

Reviewer #1: 

1. The authors state that with the small number of cases in this table from the adolescents, it was found more appropriate to quote the p-value obtained from Fisher's exact test. But in the section of statistical analysis of the revised manuscript, they didn't mention about Fisher's exact test.

Thank you for your comment. A statement addressing this oversight has now been added.

2. Did the authors consider to exclude six adolescents from GEE model and Fisher's exact test / Chi-square test? Because there were only six adolescent patients in the data, the sample size of the adolescents was small. The situation of six adolescent patients could be discussed separately.

We developed the GEE model using all the data (including the adolescents), but then re-ran the model after excluding the records belonging to these 6 patients. We found that the two models agreed closely, and that the statistical significance of the independent variables identified using the model based on all the records was very similar to those obtained after excluding the adolescents. On this basis, we felt it was justified to include the 57 off-label drug records belonging to these 6 adolescent patients.

3. The items in Table 2 are confused. Some drugs have two frequencies (e.g., paracetamol, ondansetron, ...). The cumulative frequency of age/weight, indication, dose/frequency, and special precautions for ranitidine is 243, which was not equal to the frequency of ranitidine in Table 2. It might be that the frequency of ranitidine in Table 2 was counted as combination of reasons, but the frequency in Table 3 was not. Even so, it's easy to confuse. Information about Table 2 might be addressed in the context or the footnote, but not in the title of Table 2.

We have modified Table 2 to include all of the ten most commonly prescribed drugs in Table 1. For each drug, the reason, or combination of reasons, for off-label prescribing, is shown in Table 2.

Further, we have added a line to the first column to show the number of patients who were taking each drug (in addition to the number of drug records on which each line of the table was based). This may clarify the situation. 

We reviewed Table 3, and found that there was an error in the numbers. We consider that the expansion of Table 2 now includes all the information that we wanted to describe in Table 3, and therefore we think it is best to delete Table 3. It was included only as an example, but we thank the reviewer for pointing out the inconsistency in the numbers. 

Reviewer #2: 

The manuscript has been largely improved by addressing most of the issues raised from previous review. But there is still one issue to be solved, the title does not fit the context. They are suggested to append 'during the period Aug.-Oct. 2014' to the title of manuscript to reflect the context and avoid the over-claiming issue.

Many thanks for your suggestion. The title has been changed to: ‘High prevalence of off-label and unlicensed paediatric prescribing in a hospital in Indonesia during the period Aug. – Oct. 2014’

Reviewer #3: 

The reason is still unknown why the size of study population is only 200 randomly selected patients rather than include all 307 patients? Although 200 is enough to provides a sufficient sample size for regression and related analysis to be performed, how about all 307 patients? Please show the results of non selected dataset.

Under the ethical principles related to the conduct of research, it is unethical to collect additional data than is necessary to demonstrate the objectives which were being pursued, as the data is private and confidential information. In this case, a list of a random sample of 200 medical records (from a population of 307 medical records) was prepared before patient medical records were accessed. Data was only collected from the 200 patient medical records which were randomly selected. As this was a random sample, that sample should be representative of the 307 that constituted the sampling frame. Further, a statistician was consulted to ensure the random sample was sufficient for the purpose of the analysis performed.

---

## [Decision Letter · Decision Letter 2]

27 Dec 2019

High prevalence of off-label and unlicensed paediatric prescribing in a hospital in Indonesia during the period Aug. - Oct. 2014

PONE-D-19-20113R2

Dear Dr. Czarniak,

We are pleased to inform you that your manuscript has been judged scientifically suitable for publication and will be formally accepted for publication once it complies with all outstanding technical requirements.

With kind regards,

Jinn-Moon Yang

Academic Editor

PLOS ONE

Additional Editor Comments (optional):

Reviewers' comments:

Reviewer's Responses to Questions

**Comments to the Author**

1. If the authors have adequately addressed your comments raised in a previous round of review and you feel that this manuscript is now acceptable for publication, you may indicate that here to bypass the “Comments to the Author” section, enter your conflict of interest statement in the “Confidential to Editor” section, and submit your "Accept" recommendation.

Reviewer #1: All comments have been addressed

Reviewer #2: All comments have been addressed

Reviewer #3: (No Response)

2. Is the manuscript technically sound, and do the data support the conclusions?

Reviewer #1: Yes

Reviewer #2: (No Response)

Reviewer #3: Yes

3. Has the statistical analysis been performed appropriately and rigorously? 

Reviewer #1: Yes

Reviewer #2: (No Response)

Reviewer #3: No

4. Have the authors made all data underlying the findings in their manuscript fully available?

Reviewer #1: Yes

Reviewer #2: (No Response)

Reviewer #3: No

5. Is the manuscript presented in an intelligible fashion and written in standard English?

Reviewer #1: Yes

Reviewer #2: (No Response)

Reviewer #3: Yes

6. Review Comments to the Author

Reviewer #1: The revised manuscript is well-presented and all comments have been addressed. I recommend for the acceptance of this manuscript for publication.

Reviewer #2: (No Response)

Reviewer #3: It is not convinced that the reason for using 200 random selected patients instead of whole 307 patients. It is right that the random sample is sufficient for the purpose of the analysis performed, however, it is used when the sample space is large. In this manuscript, the whole sample space is only 307, it is not too big for collecting data or calculating statistics. If the authors insist on the random sample procedure, please use resampling with the bootstrap method to show there is no significant difference among each resampling.

7. PLOS authors have the option to publish the peer review history of their article (what does this mean?). If published, this will include your full peer review and any attached files.

Reviewer #1: No

Reviewer #2: No

Reviewer #3: No

---

## [Editor Report · Acceptance letter]

31 Dec 2019

PONE-D-19-20113R2 

High prevalence of off-label and unlicensed paediatric prescribing in a hospital in Indonesia during the period Aug. - Oct. 2014 

Dear Dr. Czarniak:

I am pleased to inform you that your manuscript has been deemed suitable for publication in PLOS ONE. Congratulations! Your manuscript is now with our production department. 

With kind regards,

on behalf of

Prof. Jinn-Moon Yang 

Academic Editor

PLOS ONE